# Detecting Lexical Entailment in Context

## Abstract

Detecting entailment between words is a key task for several NLP applications. Previous work has largely focused on entailment between words out of context. We propose, instead, to address lexical entailment in context, providing exemplar sentences to ground the meaning of words considered in the entailment relation. We show that contextualized word representations constructed from existing word embeddings, and word-context similarity features lead to significant improvements over context-agnostic models on two novel entailment test sets, and also improve the state-of-the-art on the related task of detecting semantic relations in context (Shwartz and Dagan, 2015).

## 1 Introduction

Many NLP applications require detecting relations between word meanings beyond synonymy and paraphrasing. For instance, given *"Carlsen plays chess."* and the question *"Which game does Carlsen play?"*, successfully answering the question requires knowing that *chess* is a kind of *game*, or more generally, that *chess* entails *game*.

While prior work has defined lexical entailment as a relation between word types (Turney and Mohammad, 2013), we argue entailment relations are better defined when illustrating word meaning with an example in context. Ignoring context is problematic since entailment might hold between some senses of the words, but not others. Consider the word *game* in two distinct contexts:

1. The championship *game* was played in NYC.

2. The hunters were interested in the big *game*.

Given the sentence, *Carlsen is the world* chess *champion*, chess $\implies$ game as used in the first context, while chess $\not\implies$ game in the second context.

In this paper, we investigate how to represent and compare the meaning of words in context for lexical entailment. Since distributional representations for word types have proved useful to detect lexical entailment out of context in supervised settings (Baroni et al., 2012; Roller et al., 2014; Turney and Mohammad, 2013), we propose to transform context-agnostic word type representations into contextualized representations that highlight salient properties of the context (Section 3), and use these contextualized representations with a range of semantic similarity features (Section 4) to successfully detect entailment.

As we will see, these context representations significantly improve performance over context-agnostic baselines not only in English, but also between English and French words (Section 7) on two novel datasets (Section 5). We also show that our features are sensitive to word sense changes indicated by context, and adequately capture the direction of entailment relation (Section 8). Moreover, we establish a new state-of-the-art on an existing dataset that captures a broader range of semantic relations in context (Shwartz and Dagan, 2015), and show that the proposed features, induced solely from large amounts of raw text, yield systems that perform as well, or better than existing systems that require additional human annotation.

## 2 Defining Lexical Entailment in Context

We frame the task of lexical entailment in context as a binary classification task on examples consisting of a 4-tuple $(w_l, w_r, c_l, c_r)$, where $w_l$ and $w_r$ are two words, and $c_l$ and $c_r$ are sentences which

| Words $(w_l, w_r)$ | Exemplars $(c_l, c_r)$ | Does $w_l \implies w_r$ ? |
|---|---|---|
| *staff*, *stick* | $c_l$ = He walked with the help of a wooden ***staff***. <br> $c_r$ = The kid had a candied apple on a ***stick***. | Yes |
| *staff*, *body* | $c_l$ = The hospital has an excellent nursing ***staff***. <br> $c_r$ = The whole ***body*** filed out of the auditorium. | Yes |
| *staff*, *stick* | $c_l$ = The hospital has an excellent nursing ***staff***. <br> $c_r$ = The kid had a candied apple on a ***stick***. | No |

Table 1: Examples of the context-aware lexical entailment task

illustrate each word usage. The example is treated as positive if $w_l \implies w_r$, given the meaning of each word exemplified by the contexts, and negative otherwise. Table 1 provides examples.

We say that entailment holds if the meaning of $w_l$ in the context of $c_l$ is more specific than the meaning of $w_r$ in the context of $c_r$. The nature of entailment relations captured out-of-context can be broader depending on the test beds considered[1]. Zhitomirsky-Geffet and Dagan (2009) formalize lexical entailment as a substitutional relationship, which encompasses synonymy, hypernymy, some meronymy relations, and also cause-effect relations. However, we limit entailment to the specificity relation in this work to better understand the impact of context.

Note that lexical entailment in context is not textual entailment. Recognizing textual entailment (Dagan et al., 2013) and natural language inference (Bowman et al., 2015) involves detecting entailment relations between sentences, while lexical entailment is a relation between words.

## 3 Representing Words and their Contexts for Entailment

How can we construct vector representations of the meaning of target words $w_l$ and $w_r$ in their respective exemplar contexts $c_l$ and $c_r$? We start from existing representations for word types which have proven useful for detecting lexical entailment and other semantic relations out of context (Baroni et al., 2012; Kruszewski and Baroni, 2015; Vylomova et al., 2016; Turney and Mohammad, 2013).

Given an example $(w_l, w_r, c_l, c_r)$, let $\vec{w}_l$ and $\vec{w}_r$ refer to the context-agnostic representations of $w_l$ and $w_r$, and let $C_l$ and $C_r$ represent the matrices obtained by row-wise stacking of the context-

---

[1]We refer the reader to Turney and Mohammad (2013) and Shwartz et al (2017) for comprehensive surveys of supervised and unsupervised methods for the out-of-context task.

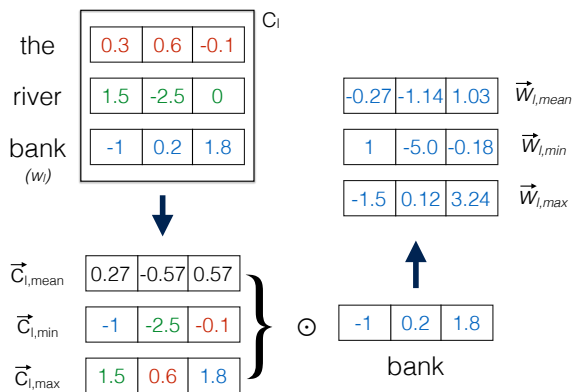

Figure 1: Constructing word-in-context representations for "bank", in the context "the river bank". $\odot$ indicates element-wise multiplication.

agnostic representations of words in $c_l$ and $c_r$ respectively.

### 3.1 Contextualized Word Representations

Following Thater et al. (2011); Erk and Padó (2008), our first approach is to apply a filter to word type representations to highlight the salient dimensions of the exemplar context, emphasizing relevant dimensions of and downplaying unimportant ones. However, while prior work represents context by averaging word vectors, we propose richer representations that better capture the salient geometrical properties of the exemplar context that might get lost by averaging (Figure 1).

First, we construct fixed length representations for the contexts $c_l$ and $c_r$ by running convolutional filters over $C_l$ and $C_r$. Specifically, we calculate the column-wise maximum, minimum and the mean over the matrices $C_l$ and $C_r$, as done by (Tang et al., 2014) for supervised sentiment classification. This yields three $d$-dimensional vectors for $c_l$ ($\vec{c}_{l,max}, \vec{c}_{l,min}, \vec{c}_{l,mean}$), and three $d$-dimensional vectors for $c_r$ ($\vec{c}_{r,max}, \vec{c}_{r,min}, \vec{c}_{r,mean}$). Computing the maximum and minimum across all vector dimensions captures the exterior surface of the "instance manifold"

(the volume in embedding space within which all words in the instance reside), while the mean summarizes the density per-dimension within the manifold (Hovy, 2015).

Second, we transform initial context-agnostic representations for target word types by taking an element-wise product of the word type vectors ($\vec{w}_*$) with vectors representing salient dimensions of the exemplar context ($\vec{c}_{*,max}, \vec{c}_{*,min}, \vec{c}_{*,mean}$) where $* \in \{l, r\}$. This yields three $d$-dimensional vectors for $w_l$ ($\vec{w}_{l,max}, \vec{w}_{l,min}, \vec{w}_{l,mean}$), and three for $w_r$ ($\vec{w}_{r,max}, \vec{w}_{r,min}, \vec{w}_{r,mean}$). We refer to our final word-in-context representations for $w_l$ and $w_r$ as $\vec{w}_{l,mask}$ and $\vec{w}_{r,mask}$ respectively, where $\vec{w}_{l,mask}$ is the concatenation of $\vec{w}_{l,max}, \vec{w}_{l,min}, \vec{w}_{l,mean}$, and $\vec{w}_{r,mask}$ is also similarly constructed.

### 3.2 A Shared Vector Space for Words and Contexts: Context2vec

An alternative approach to contextualizing word representations is to directly compare the representations of words with representations of contexts. In a single language, this can be done using Context2Vec (Melamud et al., 2016), a neural model that, given a target word and its sentential context, embeds both the word and the context in the same low-dimensional space, with the objective of having the context predict the target word via a log linear model. This model approaches the state-of-the-art on lexical substitution, sentence completion, and supervised word sense disambiguation.

## 4 Comparing Words and Contexts for Entailment

Given words and context representations described above, how can we predict entailment?

### 4.1 Supervised Logistic Regression model

Prior work on lexical entailment out of context suggests that the entailment relationship between words is a learnable function of the concatenation of their individual representations (Baroni et al., 2012; Turney and Mohammad, 2013). After concatenation, they are used as features for a logistic regression (Roller et al., 2014) or an SVM classifier (Baroni et al., 2012; Turney and Mohammad, 2013). We follow the same practice with our context-aware word representations. Intuitively, the classifier learns to weight the importance of each dimension of a word representation to detect

| Equivalence | is the same as |
|---|---|
| Entailment | is more specific than |
| Negation | is the exact opposite of |
| Alternation | is mutually exclusive with |
| Other-related | is related in some other way to |
| Independent | is not related to |

Table 2: Semantic relations in CONTEXT-PPDB. Negation and alternation are conflated into a single relation while creating the dataset.

entailment. By learning different weights for the same features describing $w_l$ or $w_r$, the classifier can detect asymmetric behavior too, as we will see in Section 8.

### 4.2 Similarity Features

We hypothesize that entailment relations hold between related words and introduce similarity features to capture this non-directional relation between words and contexts.

Given a pair of vectors $(\vec{l}, \vec{r})$, we use three similarity measures: the cosine similarity $cosine(\vec{l}, \vec{r})$, the dot product $\vec{l} \cdot \vec{r}$, and the euclidean distance $\|\vec{l} - \vec{r}\|$. The cosine similarity captures the difference in the two vectors in terms of the angle between then, the euclidean distance measures the difference in magnitude, and the dot product captures both magnitude and angle.

We apply these measures to three types of representations. Our first set of similarities is intended to directly capture the similarities between contextualized word representations (Section 3.1). We calculate pairwise similarities between $(\vec{w}_{l,max}, \vec{w}_{r,max})$, $(\vec{w}_{l,min}, \vec{w}_{r,min})$, $(\vec{w}_{l,mean}, \vec{w}_{r,mean})$, as well as between the context-agnostic representations $(\vec{w}_l, \vec{w}_r)$. Second, we add similarities based on Context2Vec representations, i.e. between each pair of $\vec{w}_{l,c2v}$, $\vec{w}_{r,c2v}$, $\vec{c}_{l,c2v}$, and $\vec{c}_{r,c2v}$. Finally, following Shwartz and Dagan (2015), we capture the similarity of the most relevant word to $w_l$ in $c_r$, that to $w_r$ in $c_l$, as well as between $c_l$ and $c_r$. We therefore add the following three similarities, as well as their pairwise products:

$$\max_{w \in c_r} \vec{w}_l \cdot \vec{w}, \ \max_{w \in c_l} \vec{w}_r \cdot \vec{w}, \ \max_{w \in c_l, w' \in c_r} \vec{w} \cdot \vec{w'}$$

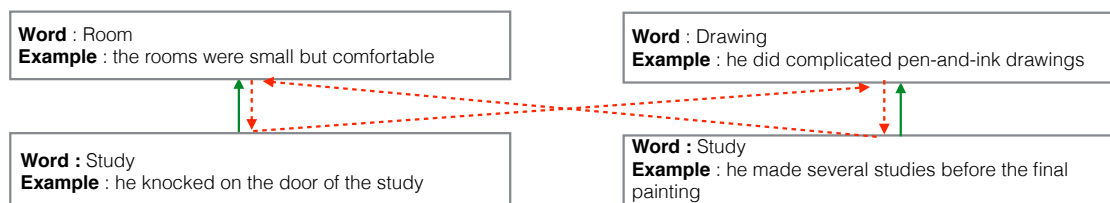

Figure 2: Sample dataset creation process based on two synsets of the word *study*. The green/solid lines indicate positive examples, while the red/dashed lines indicate negative examples

## 5 Evaluation Framework

We evaluate our models of lexical entailment in context on three complementary datasets.

### 5.1 CONTEXT-PPDB

The first dataset, CONTEXT-PPDB, is a fine-grained lexical inference dataset created using 375 word pairs from a subset of the English Paraphrase Database (Ganitkevitch et al., 2013; Pavlick et al., 2015b). These word pairs are semi-automatically labeled with semantic relations out-of-context. Shwartz and Dagan (2015) augmented them with examples of word usage in context, and re-annotating word pairs given the extra contextual information. The final dataset consists of 3750 words/contexts tuples with a corresponding label from Table 2, one of which is entailment.

### 5.2 CONTEXT-WN

In addition to CONTEXT-PPDB, we would like a test set for controlled analysis of lexical entailment in context. We would like to directly assess the sensitivity of our models to contexts that signal different word senses, as well as quantify the extent to which our models detect asymmetric entailment relations rather than semantic similarity.

Based on the above criteria, we introduce a large-scale dataset automatically extracted from WordNet (Fellbaum, 1998a). We call this dataset CONTEXT-WN. WordNet groups synonyms into *synsets*. Most synsets are further accompanied by one or more short sentences illustrating the use of the members of the synset. The idea behind CONTEXT-WN is to use these example sentences as context for the words, and hypernymy relations to draw examples of lexical entailment relations. The entire process starts from a seed list of words $W$ and proceeds as follows (see Figure 2 for an illustration) :

1. For each word type $w \in W$ with multiple synsets, obtain all synsets $S_w$.

2. For each synset $i \in S_w$, pick a hypernym synset $s_h^i$, with a corresponding word form $w_h^i$. Also obtain $c^i$ and $c_h^i$ which are example sentences corresponding to $w^i$ and $w_h^i$ respectively - $(w^i, w_h^i, c^i, c_h^i)$ serves as a positive example.

3. **Permute** the positive examples to get negative examples. From $(w^i, w_h^i, c^i, c_h^i)$ and $(w^j, w_h^j, c^j, c_h^j)$, generate negative examples $(w^i, w_h^j, c^i, c_h^j)$ and $(w^j, w_h^i, c^j, c_h^i)$.

4. **Flip** the positive examples to generate more negative examples. From $(w^i, w_h^i, c^i, c_h^i)$ generate the negative example $(w_h^i, w^i, c_h^i, c^i)$.

We run this process using the 9000 most frequent words from Wikipedia as $W$ (after filtering the top 1000 as stopwords). This yields a total of 5239 positive examples. We sample an equal number of negative examples from Step 3, with another 5239 examples being generated in Step 4. The final dataset consists of 10478 negative examples.

CONTEXT-WN satisfies our desiderata. The dataset has a very specific focus since we only pick hypernym-hyponym pairs. The negative examples generated in Steps 3 and 4 require discriminating between different word senses and entailment directions. Finally, with over 15000 examples distributed over 6000 word pairs, the dataset is almost five times as large as CONTEXT-PPDB.

We use a 70/5/25 train/dev/test split as in CONTEXT-PPDB. We also ensure that each set contains different word pairs, to avoid memorization and overfitting (Levy et al., 2015).

### 5.3 Crosslingual dataset

Our final dataset takes a cross-lingual view of lexical entailment in context. To create this dataset, we follow the same methodology that was used to create CONTEXT-PPDB, replacing word pairs from PPDB with a set of 100 English-French

word pairs labeled as positive out-of-context in the cross-lingual lexical entailment dataset of Vyas and Carpuat (2016). Each word pair $(w_l, w_r)$ consists of an English word and a French word. To add context, we extract 10 sentence pairs $(c_l, c_r)$ per word pair from the Wikipedia in the corresponding language. These thousand examples are then given to annotators on a crowdsourcing platform [2]. For each example $(w_l, w_r, c_l, c_r)$, the annotators were asked to judge whether $w_l \implies w_r$ given the contexts $c_l$ and $c_r$. Each example was annotated by 5 workers[3].

For each example, we assign the majority judgment as the correct label. 4 or more annotators agree on the label on 540 examples, and we retain only these examples as our test set. This yields a dataset of 374 positive examples and 166 negative examples.

## 6 Experimental Set-up

### 6.1 Monolingual

All experiments on CONTEXT-PPDB and CONTEXT-WN are with the default train/dev/test splits. We use 50 dimensional off-the-shelf GloVe embeddings (Pennington et al., 2014) to create $\vec{w}_l$, $\vec{w}_r$, $C_l$, and $C_r$. To obtain the Context2Vec representations $\vec{w}_{l,c2v}$, $\vec{w}_{l,c2v}$, $\vec{c}_{l,c2v}$, and $\vec{c}_{r,c2v}$, we use an existing model trained on the ukWaC corpus (Ferraresi et al., 2006) .

We use Logistic Regression as our classifier for both tasks, to allow for direct comparisons with previous work. We do not tune parameters of the classifier, except for adding class weights in the CONTEXT-WN experiments to account for the unbalanced data. As evaluation metric, we use weighted F1 score to compare against previous results, as well as to account for label imbalance in CONTEXT-WN.

### 6.2 Crosslingual

We evaluate our models on cross-lingual test set in a transfer setting by training on the training set of CONTEXT-PPDB. To represent $\vec{w}_l, \vec{w}_r$, we experiment with two different pre-trained, 200-dimensional, bilingual embeddings - BiVec (Luong et al., 2015) and BiCVM (Hermann and Blunsom, 2014), both of which have been shown to be

suitable for cross-lingual semantic tasks such as dictionary induction and document classification (Upadhyay et al., 2016). The classifier is again a Logistic Regression classifier with class weights.

## 7 Results

### 7.1 CONTEXT-PPDB

**Representation-based features only** A context-agnostic baseline (Baroni et al., 2012) based on the concatenation of the word type features for $(w_l, w_r)$ reaches F1 scores of 53.5 using Glove and 54 with Context2Vec (Table 4). The contextualized word representations and similarity features yield the best result and improves performance by 14 points. The similarity features are also effective when used in combination with the Glove or Context2Vec representations, yielding improvements of 12 to 13 points on the baseline. We also note that using $\vec{c}_{l,c2v}$ and $\vec{c}_{r,c2v}$ to add contextual information performs very poorly (F = ~60), regardless of the word type feature, indicating the superiority of our masked representations.

**Adding PPDB-specific features** CONTEXT-PPDB comes with rich information about word pairs drawn from PPDB (Pavlick et al., 2015a,b). These features include scores for likelihood of context-agnostic entailment labels, distributional similarities, and probabilities of the word pair being paraphrases, among other scores. Shwartz and Dagan (2015) establish a strong baseline of 67 F1 using these features and the most salient word/context similarities described in Section 4.2. Augmenting this system with our proposed context-aware similarities, we observe an improvement of ~2 F1 points. The contextualized representations further gives a ~3 point boost. Overall, our new context aware features help in improving upon the previous state-of-the-art by 4.8 F1 points.

### 7.2 CONTEXT-WN

We repeat all experiments on CONTEXT-WN (Table 4), except for those that require PPDB-specific features. The trends on CONTEXT-WN are similar to those on CONTEXT-PPDB. Using similarity features with the context-agnostic baseline, improves performance by ~4 F1 points. Adding the context-aware representations, further gives a tiny boost, enabling us to score 5 F1 points more than

---

[2] www.crowdflower.com

[3] We will release further details of our annotation task including guidelines and the specific task after the review period.

| Word Type Features | Context-aware Features | | Scores | | |
|---|---|---|---|---|---|
| | Similarities | Representations | P | R | F |
| ✗ | All | $[\vec{w}_{l,mask} ; \vec{w}_{r,mask}]$ | 0.674 | 0.694 | **0.677** |
| $[\vec{w}_l ; \vec{w}_r]$ | ✗ | ✗ | 0.531 | 0.539 | 0.535 |
| $[\vec{w}_l ; \vec{w}_r]$ | ✗ | $[\vec{w}_{l,mask} ; \vec{w}_{r,mask}]$ | 0.512 | 0.521 | 0.516 |
| $[\vec{w}_l ; \vec{w}_r]$ | All | ✗ | 0.642 | 0.642 | **0.642** |
| $[\vec{w}_l ; \vec{w}_r]$ | All | $[\vec{w}_{l,mask} ; \vec{w}_{r,mask}]$ | 0.638 | 0.637 | 0.637 |
| $[\vec{w}_l ; \vec{w}_r]$ | All | $[\vec{c}_{l,c2v} ; \vec{c}_{r,c2v}]$ | 0.605 | 0.604 | 0.603 |
| $[\vec{w}_{l,c2v} ; \vec{w}_{r,c2v}]$ | ✗ | ✗ | 0.533 | 0.569 | 0.540 |
| $[\vec{w}_{l,c2v} ; \vec{w}_{r,c2v}]$ | ✗ | $[\vec{w}_{l,mask} ; \vec{w}_{r,mask}]$ | 0.518 | 0.545 | 0.526 |
| $[\vec{w}_{l,c2v} ; \vec{w}_{r,c2v}]$ | All | ✗ | 0.655 | 0.674 | **0.659** |
| $[\vec{w}_{l,c2v} ; \vec{w}_{r,c2v}]$ | All | $[\vec{w}_{l,mask} ; \vec{w}_{r,mask}]$ | 0.652 | 0.670 | 0.656 |
| $[\vec{w}_{l,c2v} ; \vec{w}_{r,c2v}]$ | All | $[\vec{c}_{l,c2v} ; \vec{c}_{r,c2v}]$ | 0.601 | 0.601 | 0.600 |
| $PPDB(w_l, w_r)$ | Most salient only | ✗ | 0.677 | 0.685 | 0.670 |
| $PPDB(w_l, w_r)$ | All | ✗ | 0.695 | 0.701 | 0.692 |
| $PPDB(w_l, w_r)$ | All | $[\vec{w}_{l,mask} ; \vec{w}_{r,mask}]$ | 0.721 | 0.720 | **0.718** |

Table 3: Experimental results on CONTEXT-PPDB. The evaluation metric is weighted F score.

the baseline. Again, we observe that using $\vec{c}_{l,c2v}$ and $\vec{c}_{r,c2v}$ does worse than the alternatives.

The absolute value of gains on CONTEXT-WN is smaller than on CONTEXT-PPDB, as can be expected, given that CONTEXT-WN was designed to be challenging (Section 5.2).

### 7.3 Cross-lingual dataset

Results on the cross-lingual dataset are similar to monolingual results (Table 5). Context-aware features outperform context-agnostic baselines - similarities boost the score by 3 and 6 points in the BiVec and BiCVM settings respectively. The contextualized word representations, when used instead of the context-agnostic representations, give a boost of almost 3 points, allowing us to score 6 points and 10 points above the context-agnostic baseline in the BiVec and BiCVM cases, respectively. The strength of these results attest to the generalizable nature of our features, which helps capture correspondences beyond a single language.

## 8 Analysis

The experiments in Section 7 attest to the overall strength of our features. In this section, we aim to further test the assumptions underlying our proposed context representations.

### 8.1 Sensitivity to context

How sensitive are our models to changes in contexts? To answer this, we focus on the subset of CONTEXT-WN that comprises of positive examples, and the equivalent number of negative examples from Step 3 of the dataset creation. These examples are created by permuting the contexts of the positive examples, and thus, can directly help in answering our question. We analyze the predictions made on this subset by our model using a metric we call 'Macro-F1', defined as the weighted F1 calculated over each $(w_l, w_r)$ word pair, and then averaged over all word pairs.

Models using context-aware features do consistently better on Macro-F1 than those without (Table 4). Interestingly though, the model that uses $[\vec{c}_{l,c2v} ; \vec{c}_{r,c2v}]$ does the best on this metric, indicating that while directly using the Context2Vec representations might not be ideal, these representation do capture some useful contextual knowledge.

### 8.2 Sensitivity to Entailment Direction

To quantitatively evaluate our claim of our features learning to discriminate directionality 4, we report results on the subset of CONTEXT-WN that consists of all positive examples, and the equivalent number of flipped negative examples generated in Step 4. We measure directionality by looking at the number of example pairs $((w_l, w_r, c_l, c_r),$ $(w_r, w_l, c_r, c_l))$ where both examples are correctly labeled, i.e. the former is labeled as $\implies$ and the latter as $\not\implies$ .

Context agnostic baselines detect direction significantly above chance (Table 4). Adding our

| Word Type Feat.s | Context-aware Features | | Scores | | | | |
|---|---|---|---|---|---|---|---|
| | Similarities | Representations | P | R | F | Macro F1 | Pairwise Acc. |
| Random | | | 0.5 | 0.5 | 0.5 | 0.5 | 0.25 |
| ✗ | All | $[\vec{w}_{l,mask} ; \vec{w}_{r,mask}]$ | 0.700 | 0.661 | 0.670 | 0.677 | 0.623 |
| $[\vec{w}_l ; \vec{w}_r]$ | ✗ | ✗ | 0.653 | 0.615 | 0.625 | 0.594 | 0.582 |
| $[\vec{w}_l ; \vec{w}_r]$ | ✗ | $[\vec{w}_{l,mask} ; \vec{w}_{r,mask}]$ | 0.656 | 0.618 | 0.628 | 0.608 | 0.592 |
| $[\vec{w}_l ; \vec{w}_r]$ | All | ✗ | 0.700 | 0.663 | 0.672 | 0.651 | 0.597 |
| $[\vec{w}_l ; \vec{w}_r]$ | All | $[\vec{w}_{l,mask} ; \vec{w}_{r,mask}]$ | 0.704 | 0.666 | **0.675** | 0.673 | **0.617** |
| $[\vec{w}_l ; \vec{w}_r]$ | All | $[\vec{c}_{l,c2v} ; \vec{c}_{r,c2v}]$ | 0.692 | 0.658 | 0.667 | **0.682** | 0.592 |
| $[\vec{w}_{l,c2v} ; \vec{w}_{r,c2v}]$ | ✗ | ✗ | 0.695 | 0.655 | 0.664 | 0.663 | 0.667 |
| $[\vec{w}_{l,c2v} ; \vec{w}_{r,c2v}]$ | ✗ | $[\vec{w}_{l,mask} ; \vec{w}_{r,mask}]$ | 0.694 | 0.655 | 0.665 | 0.668 | 0.660 |
| $[\vec{w}_{l,c2v} ; \vec{w}_{r,c2v}]$ | All | ✗ | 0.728 | 0.689 | **0.697** | 0.704 | **0.700** |
| $[\vec{w}_{l,c2v} ; \vec{w}_{r,c2v}]$ | All | $[\vec{w}_{l,mask} ; \vec{w}_{r,mask}]$ | 0.724 | 0.685 | 0.694 | 0.704 | 0.691 |
| $[\vec{w}_{l,c2v} ; \vec{w}_{r,c2v}]$ | All | $[\vec{c}_{l,c2v} ; \vec{c}_{r,c2v}]$ | 0.713 | 0.678 | 0.687 | **0.708** | 0.638 |

Table 4: Results on CONTEXT-WN. Macro-F1 and Pairwise accuracy, are intended to capture context-awareness and directionality-discrimination abilities of our features, repsectively.

| Word Type Features | Context-aware Features | | Scores | | |
|---|---|---|---|---|---|
| | Similarities | Representations | P | R | F |
| Random | | | 0.570 | 0.481 | 0.501 |
| $[\vec{w}_{l,bivec} ; \vec{w}_{r,bivec}]$ | ✗ | ✗ | 0.618 | 0.537 | 0.560 |
| $[\vec{w}_{l,bivec} ; \vec{w}_{r,bivec}]$ | ✗ | $[\vec{w}_{l,mask} ; \vec{w}_{r,mask}]$ | 0.618 | 0.537 | 0.560 |
| $[\vec{w}_{l,bivec} ; \vec{w}_{r,bivec}]$ | All | ✗ | 0.632 | 0.576 | 0.595 |
| $[\vec{w}_{l,bivec} ; \vec{w}_{r,bivec}]$ | All | $[\vec{w}_{l,mask} ; \vec{w}_{r,mask}]$ | 0.620 | 0.554 | 0.575 |
| ✗ | All | $[\vec{w}_{l,mask} ; \vec{w}_{r,mask}]$ | 0.650 | 0.607 | **0.622** |
| $[\vec{w}_{l,bicvm} ; \vec{w}_{r,bicvm}]$ | ✗ | ✗ | 0.651 | 0.517 | 0.530 |
| $[\vec{w}_{l,bicvm} ; \vec{w}_{r,bicvm}]$ | ✗ | $[\vec{w}_{l,mask} ; \vec{w}_{r,mask}]$ | 0.655 | 0.528 | 0.543 |
| $[\vec{w}_{l,bicvm} ; \vec{w}_{r,bicvm}]$ | All | ✗ | 0.681 | 0.574 | 0.590 |
| $[\vec{w}_{l,bicvm} ; \vec{w}_{r,bicvm}]$ | All | $[\vec{w}_{l,mask} ; \vec{w}_{r,mask}]$ | 0.682 | 0.566 | 0.581 |
| ✗ | All | $[\vec{w}_{l,mask} ; \vec{w}_{r,mask}]$ | 0.676 | 0.614 | **0.629** |

Table 5: Results on cross-lingual dataset with training on CONTEXT-PPDB using bilingual embeddings.

context features improves by ~3, with the overall best performing model again scoring highest on this metric. It is reassuring to observe that improved detection of context with our features does not come at the cost of detecting entailment direction. Unlike in the previous section, the model that uses $[\vec{c}_{l,c2v} ; \vec{c}_{r,c2v}]$ is consistently poor, scoring lower than the corresponding context-agnostic baseline.

## 8.3 Contextualized Masks

We also hypothesized that masked contextualized representations based on the full volume of the context using $min$ and $max$ operations (Section 3.1) better capture salient context dimensions than the more usual vector averaging approach. We test this hypothesis empirically by replacing masked word-in-context representations $\vec{w}_{l,mask}$ and $\vec{w}_{r,mask}$ by two other ways to capture context. In the first method, we use the mean of the contexts $(\vec{c}_{l,mean}, \vec{c}_{r,mean})$. In the second method, we use $(\vec{w}_{l,mean}, \vec{w}_{r,mean})$, i.e. the masked representations calculated by using only the mean. On both CONTEXT-PPDB and CONTEXT-WN, our preferred method outperforms the two alternatives by 2-3 F1 points. In fact, on CONTEXT-WN we can see that our method also captures directionality best.

## 9 Related Work

**WordNet and lexical entailment** The is-a hierarchy of WordNet (Fellbaum, 1998b) is a prominent source of information for unsupervised detection of hypernymy and entailment (Harabagiu

| Dataset | Context aware features | | Scores | | | | |
|---------|------------|-----------------|-------|-------|-------|----------|--------------|
|         | Similarities | Representations | P     | R     | F     | Macro F1 | Pairwise Acc. |
| CONTEXT-PPDB | All | $[\vec{w}_{l,mean} ; \vec{w}_{r,mean}]$ | 0.646 | 0.672 | 0.643 | | |
|  | All | $[\vec{c}_{l,mean} ; \vec{c}_{r,mean}]$ | 0.655 | 0.678 | 0.652 | NA | NA |
|  | All | $[\vec{w}_{l,mask} ; \vec{w}_{r,mask}]$ | 0.674 | 0.694 | **0.677** | | |
| CONTEXT-WN | All | $[\vec{w}_{l,mean} ; \vec{w}_{r,mean}]$ | 0.674 | 0.635 | 0.645 | 0.641 | 0.577 |
|  | All | $[\vec{c}_{l,mean} ; \vec{c}_{r,mean}]$ | 0.688 | 0.648 | 0.657 | 0.665 | 0.608 |
|  | All | $[\vec{w}_{l,mask} ; \vec{w}_{r,mask}]$ | 0.700 | 0.661 | **0.670** | **0.677** | **0.623** |

Table 6: Impact of masks on CONTEXT-PPDB and CONTEXT-WN.

and Moldovan, 1998; Shwartz et al., 2015), as well as a source of various datasets (Baroni and Lenci, 2011; Baroni et al., 2012). The dataset we introduce in this work is inspired by the latter line of work, but instead of just extracting word pairs we also obtain exemplar contexts from WordNet.

**Modeling word meaning in context** Several approaches have been proposed to model the meaning of a word in a given context to capture semantic equivalence in tasks such as lexical substitution, word sense disambiguation or paraphrase ranking (but not entailment).

One line of work (Dinu and Lapata, 2010; Reisinger and Mooney, 2010) treats each word as a set of latent word senses. These models methods start with token representations for individual occurrences of a word and then choose a set of token vectors based on the current context. An alternate set of models (Erk and Padó, 2008; Thater et al., 2011; Dinu et al., 2012) avoids defining a fixed set of word senses, and instead contextualizes word type vectors as we do in this paper. These models share the idea of using an element-wise multiplication to apply a context mask to word type representations. The nature of the context representation varies: Erk and Pado (2008) use inverse selectional preferences; Thater et al (2010) combine a first order representation for the context with a second order representation for the target, Thater et al. (2011) rely on syntactic dependencies to define context. Apidianaki (2016) shows that bag-of-word context representation within a small context window works as well as syntactic definitions of context for ranking paraphrases in context.

The use of convolution is motivated by success of similar models on sentence classification tasks. Tang et al (2014) uses convolution over embedding matrices for unigrams, bigrams, and trigrams, while Hovy (2015) uses just unigrams. However, all these works only use convolved representations

to predict properties of the sentence (e.g., sentiment). We use them, instead, to contextualize our target word representations.

**In-context lexical semantic tasks** Besides entailment, other lexical semantic tasks studied in the presence of context include lexical substitution (McCarthy and Navigli, 2007), cross-lingual lexical substitution (Mihalcea et al., 2010) and paraphrase ranking (Apidianaki, 2016). The last work is also notable because of their successful use of models of word-meaning in context from Thater et al (2011), which is closely related to our methods.

## 10 Conclusion

We proposed to address lexical entailment in context, providing exemplar sentences to ground the meaning of words being considered for entailment. We show that contextualized word representations constructed by transforming context-agnostic representations, combined with word-context similarity features, lead to large improvements over context-agnostic model, not only in English, but also between English and French words on two novel datasets. We also improve the state-of-the-art on a related task of detecting semantic relations in context. Our features are sensitive to changes in entailment based on context, and also capture the directionality of entailment.

In future work, we aim to further improve performance on both monolingual and cross-lingual datasets. We anticipate that richer features, capturing second-order comparisons used to detect lexical contrast (Mohammad et al., 2013) and entailment out of context (Turney and Mohammad, 2013), might be useful for this purpose, perhaps in combination with non-linear classifiers. Given the usefulness of similarities, we also plan to investigate asymmetric scoring functions(Kotlerman et al., 2010; Shwartz et al., 2017).

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
