# Peer review of "Detecting Lexical Entailment in Context"

_ACL 2017 — decision unknown_

[Official Review · Reviewer 1 · rating 4 · confidence 2]
soundness 3 · originality 4 · clarity 4 · impact 3 · substance 4 · appropriateness 5 · meaningful comparison 4 · presentation format Oral Presentation

- Strengths: A well written paper, examining the use of context in lexical
entailment task is a great idea, a well defined approach and experimental
set-up and good analysis of the results 

- Weaknesses: Some information is missing or insufficient, e.g., the table
captions should be more descriptive, a clear description for each of the word
type features should be given.

General Discussion: 

The paper presents a proposal of consideration of context
in lexical entailment task. The results from the experiments demonstrate that
context-informed models do better than context-agnostic models on the
entailment task. 

I liked the idea of creating negative examples to get negative annotations
automatically in the two ways described in the paper based on WordNet positive
examples. (new dataset; an interesting method to develop dataset)

I also liked the idea of transforming already-used context-agnostic
representations into contextualized representations, experimenting with
different ways to get contextualized representations (i.e., mask vs
contetx2vec), and testing the model on 3 different datasets (generalizability
not just across different datasets but also cross-linguistically).

Motivations for various decisions in the experimental design were good to
see, e.g., why authors used the split they used for CONTEXT-PPDB (it showed
that they thought out clearly what exactly they were doing and why).

Lines 431-434: authors might want to state briefly how the class weights were
determined and added to account for the unbalanced data in the CONTEXT-WN
experiments. Would it affect direct comparisons with previous work, in what
ways? 

Change in Line 589: directionality 4 --> directionality, as in Table 4

Suggested change in Line 696-697: is-a hierarchy of WordNet --> "is-a"
hierarchy of WordNet 

For the sake of completeness, represent "mask" also in Figure 1.

I have read the author response.

[Official Review · Reviewer 2 · rating 2 · confidence 5]
soundness 3 · originality 4 · clarity 3 · impact 3 · substance 1 · appropriateness 5 · meaningful comparison 4 · presentation format Poster

This paper proposes a method for recognizing lexical entailment (specifically,
hypernymy) in context. The proposed method represents each context by
averaging, min-pooling, and max-pooling its word embeddings. These
representations are combined with the target word's embedding via element-wise
multiplication. The in-context representation of the left-hand-side argument is
concatenated to that of the right-hand-side argument's, creating a single
vectorial representation of the input. This input is then fed into a logistic
regression classifier.

In my view, the paper has two major weaknesses. First, the classification model
used in this paper (concat + linear classifier) was shown to be inherently
unable to learn relations in "Do Supervised Distributional Methods Really Learn
Lexical Inference Relations?" (Levy et al., 2015). Second, the paper makes
superiority claims in the text that are simply not substantiated in the
quantitative results. In addition, there are several clarity and experiment
setup issues that give an overall feeling that the paper is still half-baked.

= Classification Model =

Concatenating two word vectors as input for a linear classifier was
mathematically proven to be incapable of learning a relation between words
(Levy et al., 2015). What is the motivation behind using this model in the
contextual setting?

While this handicap might be somewhat mitigated by adding similarity features,
all these features are symmetric (including the Euclidean distance, since |L-R|
= |R-L|). Why do we expect these features to detect entailment?

I am not convinced that this is a reasonable classification model for the task.

= Superiority Claims =

The authors claim that their contextual representation is superior to
context2vec. This is not evident from the paper, because:

1) The best result (F1) in both table 3 and table 4 (excluding PPDB features)
is the 7th row. To my understanding, this variant does not use the proposed
contextual representation; in fact, it uses the context2vec representation for
the word type.

2) This experiment uses ready-made embeddings (GloVe) and parameters
(context2vec) that were tuned on completely different datasets with very
different sizes. Comparing the two is empirically flawed, and probably biased
towards the method using GloVe (which was a trained on a much larger corpus).

In addition, it seems that the biggest boost in performance comes from adding
similarity features and not from the proposed context representation. This is
not discussed.

= Miscellaneous Comments =

- I liked the WordNet dataset - using the example sentences is a nice trick.

- I don’t quite understand why the task of cross-lingual lexical entailment
is interesting or even reasonable.

- Some basic baselines are really missing. Instead of the "random" baseline,
how well does the "all true" baseline perform? What about the context-agnostic
symmetric cosine similarity of the two target words?

- In general, the tables are very difficult to read. The caption should make
the tables self-explanatory. Also, it is unclear what each variant means;
perhaps a more precise description (in text) of each variant could help the
reader understand?

- What are the PPDB-specific features? This is really unclear.

- I could not understand 8.1.

- Table 4 is overfull.

- In table 4, the F1 of "random" should be 0.25.

- Typo in line 462: should be "Table 3"

= Author Response =

Thank you for addressing my comments. Unfortunately, there are still some
standing issues that prevent me from accepting this paper:

- The problem I see with the base model is not that it is learning prototypical
hypernyms, but that it's mathematically not able to learn a relation.

- It appears that we have a different reading of tables 3 and 4. Maybe this is
a clarity issue, but it prevents me from understanding how the claim that
contextual representations substantially improve performance is supported.
Furthermore, it seems like other factors (e.g. similarity features) have a
greater effect.

[Official Review · Reviewer 3 · rating 2 · confidence 4]
soundness 3 · originality 4 · clarity 2 · impact 3 · substance 3 · appropriateness 5 · meaningful comparison 4 · presentation format Poster

This paper addresses the task of lexical entailment detection in context, e.g.
is "chess" a kind of "game" given a sentence containing each of the words --
relevant for QA. The major contributions are:

(1) a new dataset derived from WordNet using synset exemplar sentences, and 

(2) a "context relevance mask" for a word vector, accomplished by elementwise
multiplication with feature vectors derived from the context sentence. Fed to a
logistic regression classifier, the masked word vectors just beat state of the
art on entailment prediction on a PPDB-derived dataset from previous
literature. Combined with other existing features, they beat state of the art
by a few points. They also beats the baseline on the new WN-derived dataset,
although the best-scoring method on that dataset doesn't use the masked
representations.

The paper also introduces some simple word similarity features (cosine,
euclidean distance) which accompany other cross-context similarity features
from previous literature. All of the similarity features, together, improve the
classification results by a large amount, but the features in the present paper
are a relatively small contribution.

The task is interesting, and the work seems to be correct as far as it goes,
but incremental. The method of producing the mask vectors is taken from
existing literature on encoding variable-length sequences into min/max/mean
vectors, but I don't think they've been used as masks before, so this is novel.
However, excluding the PPDB features it looks like the best result does not use
the representation introduced in the paper.

A few more specific points:

In the creation of the new Context-WN dataset, are there a lot of false
negatives resulting from similar synsets in the "permuted" examples? If you
take word w, with synsets i and j, is it guaranteed that the exemplar context
for a hypernym synset of j is a bad entailment context for i? What if i and j
are semantically close?

Why does the masked representation hurt classification with the
context-agnostic word vectors (rows 3, 5 in Table 3) when row 1 does so well?
Wouldn't the classifier learn to ignore the context-agnostic features?

The paper should make clearer which similarity measures are new and which are
from previous literature. It currently says that previous lit used the "most
salient" similarity features, but that's not informative to the reader.

The paper should be clearer about the contribution of the masked vectors vs the
similarity features. It seems like similarity is doing most of the work.

I don't understand the intuition behind the Macro-F1 measure, or how it relates
to "how sensitive are our models to changes in context" -- what changes? How do
we expect Macro-F1 to compare with F1?

The cross-language task is not well motivated.

Missing a relevant citation: Learning to Distinguish Hypernyms and Co-Hyponyms.
Julie Weeds, Daoud Clarke, Jeremy Reffin, David Weir and Bill Keller. COLING
2014.

==

I have read the author response. As noted in the original reviews, a quick
examination of the tables shows that the similarity features make the largest
contribution to the improvement in F-score on the two datasets (aside from PPDB
features). The author response makes the point that similarities include
contextualized representations. However, the similarity features are a mixed
bag, including both contextualized and non-contextualized representations. This
would need to be teased out more (as acknowledged in the response).

Neither Table 3 nor 4 gives results using only the masked representations
without the similarity features. This makes the contribution of the masked
representations difficult to isolate.